# Comparative Analysis of Stroke, Marital Intimacy, Marital Satisfaction and Divorce Intention According to the Type of Participation in Marital Leisure Sports Activities

**DOI:** 10.3390/bs14090757

**Published:** 2024-08-27

**Authors:** Ji-Hye Yang, Hye Jin Yang, Si Cheol Jung, Chulhwan Choi, Chul-Ho Bum

**Affiliations:** 1Department of Physical Education, Graduate School, Kyung Hee University, Seocheon-dong 1, Giheung-gu, Yongin-si 17104, Gyeonggi-do, Republic of Korea; didwlgp9@khu.ac.kr (J.-H.Y.); y0108577@khu.ac.kr (H.J.Y.); jsc6277@naver.com (S.C.J.); 2Department of Physical Education, Gachon University, 1342 Seongnamdaero, Sujeong-gu, Seongnam-si 13120, Gyeonggi-do, Republic of Korea; 3Department of Golf Industry, College of Physical Education, Kyung Hee University, Seocheon-dong 1, Giheung-gu, Yongin-si 17104, Gyeonggi-do, Republic of Korea

**Keywords:** stroke, marital intimacy, marital satisfaction, divorce intention, leisure sports

## Abstract

This study examined strokes, marital intimacy, marital satisfaction, and divorce intentions among participants who are members of a couple, based on their leisure sports participation. We conducted a questionnaire survey with 321 participants. The validity and reliability were checked, and a multivariate analysis of variance was conducted to test for differences between groups. The results revealed that among couples engaging in the same leisure sports activity, positive stroke was high, and negative and no strokes were low. Couples not engaging in the same activity exhibited a partial positive effect when both or one of the individuals engaged in leisure sports activities. Among couples in which both individuals did not engage in leisure sports activities, negative results were found for all factors. Furthermore, couples engaging in the same activity showed high marital intimacy and satisfaction. Moreover, couples engaging in the same activity showed the lowest intention to divorce, whereas couples who did not engage in leisure sports activities showed the highest intention to divorce. These findings suggest that leisure sports activities positively affect relationships, and this effect increases when couples participate in the same sport. Future research should investigate methods for couples to engage in leisure activities and how they can be activated.

## 1. Introduction

In Western countries, couples have long tended to recognize each other’s values equally [1], whereas in Asian countries, unequal marriages are quite common, as men generally tend to take the lead in patriarchal cultures [2]. However, in modern society, women are recognized for their ability and value, like men. Ghuman et al. found that women’s autonomy is increasing rapidly regardless of men’s intentions [3]. In addition, since men are no longer the exclusive heads of households, given the emergence of dual-income couples and even male housewives [4], gender no longer has a significant effect on work and family. According to Schoen et al., gender equality in modern times has led to a significantly higher rate of consensual divorce than in the past [5]. Cohen suggested that young women propose divorce first [6]. Additionally, the provision of education and medical services currently supported by the state reduces the burden on individuals; therefore, couples do not have to combine their economic power by continuing their marriage, which tends to make divorce decisions easier [7]. In other words, while the influence of economic and social factors on divorce decisions among couples is insufficient, psychological factors such as marital conflict resolution have become more important [8]. Moreover, to facilitate a healthy and happy marriage, couples need to understand each other’s dissatisfaction [9], and the time couples spend together is very precious [10].

There are various activities in which couples share time and space; however, studies have reported that joint physical and leisure sports activities are excellent for maintaining and improving relationships [11]. The choice of leisure sports activity depends on one’s pursuit of physical activity and the social environment [12]. Marital leisure activities may be conducted individually without one’s spouse, together as a couple, or both [13]. Leisure activities performed together at the beginning of marriage greatly help couples in getting to know each other, but this may not be sustained because of various variables [14]. In this regard, Gsellmeier et al. stated that women who perform a lot of parenting and housework prefer performing leisure activities with new people or acquaintances rather than with family [15]. Men who work tend to spend leisure time with their family, but they also have time to rest alone or engage in leisure activities for health purposes. This seems to be a characteristic of enjoying leisure time in an environment completely different from the one in which they usually live, regardless of sex. Despite these different pursuits of leisure activities, couples attempt to participate in leisure activities together [16] because similar pursuits of leisure help couples communicate and experience enjoyment [17], increasing marital satisfaction and decreasing divorce intentions [18]. For example, Higgins et al. have reported cases in which couples have been positively influenced by their interactions through leisure [11]; hence, it is also true that they should aim for better interaction.

Transactional analysis (TA) was invented by Eric Berne in 1958 [19] as group therapy [20], and this analysis proposed to find three ego states’ (parent, adult, and child) problems [21]. Massey [22] said that it can be considered in both social psychiatry and psychology. For example, some studies already focused on TA in family or couple relationships [23,24]. Thus, it is used to examine relationships and improve communication between couples, creating an opportunity for couples to better understand each other before and after marriage [25]. In addition, TA-derived stroke is a comprehensive concept of communication that entails the language, attitude, and behavior of the other party [26]. Verbal and nonverbal communication has already become an important factor in relationship exchanges from the past; it is therefore essential for an ideal marriage [27,28]. Smooth communication increases maturity in dealing with individual emotions, which develops into empathy in understanding the other person [29]. The higher the level of consensus, the higher the intimacy of the relationship. According to White et al., intimacy is based on a high-quality marriage and consists of four dimensions: relationship orientation, commitment, care, and attention [30]. In this regard, it is important to avoid hiding each other’s feelings, because intimacy requires mutual trust, self-conditioning, and information disclosure [31]. Marital satisfaction and divorce intention are formed by a variety of factors, and according to Hirschberger et al., divorce intention can be predicted according to the degree of marital satisfaction [32]. Kelly and Conley measured marital satisfaction in terms of personality, initial social environment, marital attitude, and personality factors [33], while Holman and Jacquart [34] suggested a way to increase marital satisfaction by analyzing it in relation to leisure activities.

The divorce problem, which peaked in the late 1980s after the passage of the Liability Divorce Act, persists [35]. Modern divorce is caused by the accumulation of daily stress in marriage [36], and people who have experienced divorce are more likely to have negative experiences beyond psychological stress, such as social stress and depression [37,38]. Therefore, leisure activities that help build a healthy life and relieve stress are expected to have a positive effect on reducing divorce rates [39]. Regarding relevant prior research, Craig and Brown analyzed the joint leisure activities of couples according to the presence or absence of children [40], and Johnson et al. explored the effects of participation in joint leisure activities, leisure time, and leisure satisfaction on couples’ marital satisfaction [13]. However, studies that categorized participants into leisure activity types, as in this study, are insufficient; this highlights the importance and necessity of this study. In addition, it is true that to maintain a positive marital relationship, both spouses must feel happy in the marriage [41]. However, no matter how well-adjusted a couple is, it is necessary to have some time alone [42]; thus, it cannot be concluded that companion leisure activities are the most effective.

Therefore, based on the changing social environment and associated marital life [43], this study proposes a way to maintain better leisure activities by comparing and analyzing factors such as marital stroke, intimacy, marital satisfaction, and divorce intention. The hypothesis established in this study is as follows:

**Hypothesis** **1:**
*There will be differences in marital stroke, marital intimacy, marital satisfaction, and divorce intention between couples depending on the type of leisure activity.*


### Literature Review

Stroke is one of the components of TA theory. First developed by Berne [19], the theory was revised in various ways by the International Transactional Analysis Association [44] and began to be used to improve modern marital relationships [45]. Shirai defined a stroke as any action of recognizing and accepting a partner that is used verbally and nonverbally in human relationships [46,47]. Arora stated that humans want to receive negative strokes rather than no strokes and positive strokes rather than negative strokes [48]. Song and Kim emphasized the importance of positive strokes received from others, as they serve as an opportunity to recognize their existence again, which helps develop relationships [49]. However, compared to previous studies using TA [50,51], studies related to single-factor strokes and marital relationships are insufficient.

Cordova and Scott referred to intimacy as the process of developing human relationships [52]. In a study by Ben-Ari and Lavee, the intimacy participants felt in human relationships was largely the intimacy of friendship, emotion, and thought sharing or caring [53]. Jamieson stated that modern couples prefer the intimacy of sharing and empathizing with each other [54]. Intimacy also has a positive effect on marriage; for example, White et al. stated that intimacy is the basis of a happy marriage, and Cordova emphasized the importance of intimacy in maintaining marriage [29]. In addition, Greeff and Malherbe showed that intimacy and marital satisfaction are correlated [55], while Vannoy demonstrated that intimacy and divorce intention are correlated [56]. This shows that marital intimacy can positively or negatively affect happiness and quality of life [57].

Marital satisfaction is the level of happiness in one’s marriage [58], and divorce intention is an individual’s intention to end their marriage. Both are subjective and have a common characteristic of being influenced by the environment. Sandhya found that in Eastern countries, marital happiness tends to be found in social relationships compared to Western countries, and Indian couples do not have good intimacy with each other and underestimate the ability of family men even though they are happy in their marriage [59]. In addition, Ward et al. found that companion leisure activities that consumed significant money and time negatively affected marital satisfaction, and this was due to the couple’s different pursuits of leisure time [60]. These studies suggested important ways to increase marital satisfaction and lower divorce intention.

## 2. Materials and Methods

### 2.1. Study Participants and Data Collection

This study was conducted on 350 participants who are members of a couple engaged in leisure sports activities in Korea. Among the non-probability sampling methods, convenience sampling was used, and voluntary oral consent was obtained from all participants. From 1 February to 15 March 2024, an online survey was conducted using a self-assessment method. Subsequently, 321 responses were used for analysis, after excluding 29 incomplete responses. The groups were divided into Group 1 (couples who play the same leisure sports together), Group 2 (couples who play different sports separately), Group 3 (couples where only one of them plays leisure sports), and Group 4 (couples who do not play leisure sports). Demographic characteristics (sex, age, marriage period, income type, leisure activity type, leisure activity composition, and number of participants) are shown in Table 1.

### 2.2. Instruments

The stroke scale was used by revising and supplementing a questionnaire developed by Kim and standardized by the Korea Exchange and Analysis Association according to the purpose of this study [61]. The sub-factors consisted of 20 items, including four items on negative strokes one gives to their spouse, four items on negative strokes one receives from their spouse, four items on positive strokes one receives from their spouse, four items on positive strokes one gives to their spouse, and four items on the absence of strokes. Marital intimacy was a single factor used by Sternberg [62] and Kim [63] and was modified to fit this study, consisting of five items. Marital satisfaction, a single factor used by Kim [64] and modified to fit this study, consists of five items. Lee developed the divorce intention measure [65], and the factors revised and reconstructed by Cha were used in this study [66]. The questionnaire consisted of six items rated on a 5-point Likert scale ranging from 1 (“not at all”) to 5 (“very much”). Detailed information on the items is presented in Table 2.

### 2.3. Data Analysis

Data were statistically analyzed using SPSS 28.0. First, descriptive statistics were used to verify participants’ sociodemographic information. Next, to test the validity of the data collected in this study, an exploratory factor analysis (EFA) for the stroke factor, including the five sub-factors, was performed. Cronbach’s alpha coefficients were calculated to assess the reliability of the collected data. Finally, multivariate analysis of variance (MANOVA) and post-hoc analyses were performed to ascertain statistically significant differences in dependent variables (i.e., stroke, marital intimacy, marital satisfaction, and divorce intention) among four groups.

## 3. Results

### 3.1. Validity and Reliability

The EFA using principal component analysis was performed on a dependent variable (the stroke factor) composed of five sub-factors: (a) positive strokes I receive from my spouse (four items), (b) negative strokes I receive from my spouse (four items), (c) no stroke (four items), (d) negative strokes I give to my spouse (four items), and positive strokes I give to my spouse (four items). The remaining factors (marital intimacy, marital satisfaction, and divorce intention) were excluded from the EFA because they were single-scale factors that did not contain any sub-factors.

From the stroke factor structure, the Kaiser–Meyer–Olkin (KMO) measure showed the acceptable sample adequacy (0.898), which was greater than 0.80 [67]. Bartlett’s test of sphericity was statistically significant (*χ*^2^ = 5658.362, *df* = 190, *p* < 0.01). The EFA retained five factors, explained 81.590% of the total variance, and yielded eigenvalues greater than one and a factor structure coefficient greater than 0 0.40.

In addition, the factors indicated satisfactory Cronbach’s alpha coefficients [68] for instrument reliability greater than 0.70: (a) positive strokes I receive from my spouse (α = 0.966), (b) negative strokes I receive from my spouse (α = 0.916), (c) no stroke (α = 0.917), (d) negative strokes I give to my spouse (α = 0.905), and (e) positive strokes I give to my spouse (α = 0.896).

### 3.2. Multivariate Analysis of Variance for the Comparative Analysis

A MANOVA was performed to test the differences in (a) stroke, (b) marital intimacy, (c) marital satisfaction, and (d) divorce intention (Table 3). First, the homogeneity of covariance was verified (Box’s *M* = 322.129, *F* = 2.849, *p* < 0.001). Statistically significant differences were found among the four groups (Wilks’ lambda = 0.464, *F* = 11.364, *p* < 0.001, partial *η*^2^ = 0.143). Specifically, statistically significant differences based on mean scores were revealed for all eight dependent variables: (a) stroke (including five sub-factors), (b) marital intimacy, (c) marital satisfaction, and (d) divorce intention. Detailed results of the MANOVA are presented in Table 3.

Next, as this study analyzed statistical differences between four groups, additional post-hoc analyses were mandatory to determine where statistically significant differences existed among the groups. Based on the results of post-hoc analyses, first, respondents in Groups 1 and 2 reported higher mean scores than those in Groups 3 and 4 on the first stroke sub-factor (positive strokes I give to my spouse). Second, respondents in Group 1 reported higher mean scores than those in Groups 2, 3, and 4 on the second stroke sub-factor (positive strokes I receive from my spouse), marital intimacy, and marital satisfaction. Third, respondents in Groups 1, 2, and 3 reported lower mean scores than those in Group 4 on the third stroke sub-factor (negative strokes I give to my spouse). Fourth, respondents in Group 1 reported lower mean scores than those in Groups 2, 3, and 4 for the fourth and fifth stroke sub-factors (negative strokes I receive from my spouse and no stroke). Finally, the survey respondents in Group 1 produced lower results than those in Groups 2 and 3, while Groups 2 and 3 produced lower results than those in Group 4. The detailed results of the post-hoc analyses are presented in Table 3 and Table 4. Additionally, more information on the mean scores of the dependent variables by the four groups is presented in Table 5.

## 4. Discussion

This study conducted a comparative analysis of marital stroke, marital intimacy, marital satisfaction, and divorce intention according to the leisure activities of people who are members of a couple, to suggest better plans for leisure sports activities and marital life. The results indicated a significant difference in stroke based on couples’ engagement in leisure sports activities. Positive strokes (both giving and receiving) were higher among couples who engaged in the same leisure sports activity (Group 1) and couples where only one spouse engaged in a leisure sports activity (Group 3) than among couples who did not engage in leisure sports activities (Group 4). Positive strokes were particularly higher in Group 1 than in the other three groups. Conversely, negative strokes were higher among couples who did not engage in leisure sports activities (Group 4) than among couples in the other three groups, and negative or no strokes were lowest in Group 1. This means that couples who perform the same sports and leisure activities together have a higher tendency to exchange positive strokes and a lower tendency to exchange negative strokes. Leisure activities are activities that can be enjoyed by oneself, free from work or daily stress [69]; hence, it can be interpreted that the positive energy of life [70] increases through leisure activities. In addition, leisure activities occupy a large part of daily life [71]; therefore, when couples willingly engage in the same leisure activities, they spend precious time together, share valuable experiences, and have more opportunities to communicate [72,73].

On the other hand, when only one or neither spouse engages in leisure sports activities, the rate of positive strokes is low and that of negative or no strokes is high. Negative strokes toward a spouse are expressed as indifference [74], which can also be attributed to the lack of communication. Studies have also reported that poor communication is a decisive factor in divorce [75]. Ward et al. showed that even if both couples live their leisure lives, activities without interaction do not promote communication [60]. Johnson et al. reported that higher participation satisfaction is more important for couples than the number of times they participate in leisure activities [13], showing that exchange activities such as exploring, sharing, and encouraging each other are important [60], and it seems that such interactions can increase positive strokes. In this study, focus was placed on couples who participate in active leisure sports activities. Kim et al. reported that active leisure activities positively affect leisure satisfaction [76]. Stroke is a concept that encompasses communication behavior [26], and positive strokes are composed of various aspects, such as physical and conditional, and not just verbal aspects such as praise. Therefore, due to the nature of sports, the relationship improves as the rate of positive strokes increases by interacting with each other through the process of physical contact and problem solving [11,77].

In this study, like the stroke variable, couples who engaged in the same leisure sports activities had higher marital intimacy and satisfaction levels than those of the other three groups. Performing the same leisure activity together provides an opportunity for couples to increase intimacy in their relationship and share a common interest, thus helping build a bond [78]. In addition, interest in each other increases by learning how to be together through leisure activities, which increases the solidarity of marital relationships [79,80,81]. Moreover, such marital intimacy is highly correlated with marital satisfaction [49]; marital satisfaction increases as marital intimacy increases. Conversely, in the case of couples who engage in separate leisure activities, studies have reported that marital satisfaction decreases [42,82,83]. One of the reasons why couple interactions in leisure activities affect marital intimacy and satisfaction is that engaging in leisure activities together influences leisure satisfaction because of the reaction or support of spouses, and this is because humans are social animals. Culturally, in Korea, communal living is considered highly important. The improvement of marital satisfaction can be further maximized through sports activities, which increase life satisfaction and physical health, as there is continuous verbal and nonverbal communication with the other party. There are also opportunities for bonding, venting energy, as well as pleasure and healing [84,85]. Previous studies, such as those by Memar Bahabadi et al. [86] and Agate et al. [77], also reported that regular family participation in sports activities increased life satisfaction, which was correlated with marital satisfaction.

However, when couples do not perform leisure sports activities together, dissatisfaction occurs, which can be linked to divorce intention. As shown in this study, the group in which both couples did not engage in leisure sports activities had a higher divorce intention than the other groups. In terms of factors other than divorce intention, couples who did not engage in leisure sports activities fared negatively. Leisure sports activity is known to improve mental health, optimistic attitudes, emotion regulation, and stress relief [87,88]. When couples do not engage in leisure sports activities, it seems difficult to positively resolve the inevitable conflict and stress between them [89]. In addition, a lower intention to divorce can be attributed to the fact that couples who engage in leisure sports activities naturally increase their interest in communication opportunities, physical contact, and emotional communication, and improve their relationships through forming new emotions.

The significance of the results was confirmed by comparing and analyzing stroke, marital intimacy, marital satisfaction, and divorce intention according to the leisure sports activity type. According to Meunier and Baker [90], it is an undeniable fact that couples with a good relationship feel happy and value the time spent together. Furthermore, it is also well known that individuals who are part of a couple with a good relationship engage in leisure activities together more frequently than those who do not have a good relationship. Based on the situation, this study found additional information that doing leisure activities together or doing another type of leisure activity respectively associated with their psychological factors. Consequently, the result of this study could help individuals who are part of a couple to improve their quality of life and happiness in married life.

## 5. Conclusions and Future Research

This study compared and analyzed marital stroke, marital intimacy, marital satisfaction, and divorce intention according to the type of marital leisure sports activity, and the conclusions derived are as follows. First, when couples engage in the same leisure sports activity, positive stroke is high, and negative or no strokes are low. Although couples were not the same, there was a partial positive effect when each of them or only one of them engaged in leisure sports activities; when both couples did not engage in leisure sports activities, negative results were found for all factors. Second, couples who engaged in the same leisure sports activity showed high marital intimacy and satisfaction. Third, couples who engaged in the same leisure sports activity showed the lowest intention to divorce, whereas couples who did not engage in leisure sports activities showed the highest intention to divorce.

Clearly, sharing leisure sports activity is an important factor in increasing marital satisfaction and reducing divorce intention, which are important factors in marital relationships. In addition, leisure sports activities had a partially positive effect on marital relationships. In contrast, if only one or neither spouse engaged in leisure sports activities, most of the factors showed negative results; therefore, it was judged that couples needed to engage in the same leisure sports activities for a healthy marital relationship.

Based on the results of this study, the following suggestions are made for meaningful follow-up studies: First, this study focused on the presence or absence of sports and leisure activities, but it is judged that in follow-up research, types of leisure activities should be subdivided. Second, this study included 321 participants. However, to generalize the results of this study, it is necessary to increase the number of participants through random sampling. Third, follow-up studies are needed to develop programs that can increase couple leisure sports activities or prepare effective alternatives.

## Figures and Tables

**Table 1 behavsci-14-00757-t001:** Sociodemographic information of survey respondents by groups.

		Group 1	Group 2	Group 3	Group 4
Gender	Male	34 (39.1%)	49 (57.6%)	39 (55.7%)	36 (45.6%)
	Female	53 (60.9%)	36 (42.4%)	31 (44.3%)	43 (54.4%)
Age	20s	8 (9.2%)	11 (12.9%)	8 (11.4%)	14 (17.7%)
	30s	30 (34.5%)	16 (18.8%)	16 (22.9%)	16 (20.3%)
	40s	16 (18.4%)	13 (15.3%)	22 (31.4%)	23 (29.1%)
	50s	24 (27.6%)	28 (32.9%)	17 (24.3%)	16 (20.3%)
	Over 60s	9 (10.3%)	17 (20.0%)	7 (10.0%)	10 (12.7%)
Marriage duration	Less than 10 yrs	42 (48.3%)	32 (37.6%)	29 (41.4%)	40 (50.6%)
	Less than 20 yrs	10 (11.5%)	18 (21.2%)	22 (31.4%)	16 (20.3%)
	Less than 30 yrs	26 (29.9%)	19 (22.4%)	13 (18.6%)	12 (15.2%)
	Less than 40 yrs	8 (9.2%)	15 (17.6%)	6 (8.6%)	11 (13.9%)
	Unknown	1 (1.1%)	1 (1.2%)	-	-
Family income	Single	33 (37.9%)	31 (36.5%)	32 (45.7%)	33 (41.8%)
	Dual	50 (57.5%)	50 (58.8%)	37 (52.9%)	43 (54.4%)
	Unknown	4 (4.6%)	4 (4.7%)	1 (1.4%)	3 (3.8%)
Leisure activity duration	None	-	-	8 (11.4%)	63 (79.9%)
	Less than 1 yr	14 (16.1%)	23 (27.1%)	22 (31.4%)	11 (13.9%)
	1–5 yrs	33 (16.1%)	32 (37.6%)	9 (12.9%)	3 (3.8%)
	5–10 yrs	11 (12.6%)	6 (7.1%)	14 (20.0%)	-
	10–20 yrs	21 (24.1%)	17 (20.0%)	9 (12.9%)	2 (2.5%)
	Over 20 yrs	7 (8.0%)	6 (7.1%)	8 (11.4%)	-
	Unknown	1 (1.1%)	1 (1.2%)	-	-
Frequency of leisure activity	None	-	-	13 (18.6%)	70 (88.6%)
	Once a month	10 (11.5%)	17 (20.0%)	13 (18.6%)	3 (3.8%)
	2–3 times a month	34 (39.1%)	31 (36.5%)	22 (31.4%)	2 (2.5%)
	Once a week	16 (18.4%)	20 (23.5%)	9 (12.9%)	1 (1.3%)
	More than 2 a week	22 (25.3%)	15 (17.6%)	7 (10.0%)	3 (3.8%)
	Almost everyday	4 (4.6%)	2 (2.4%)	6 (8.6%)	-
	Unknown	1 (1.1%)	-	-	-
	Total	87 (100.0%)	85 (100.0%)	70 (100.0%)	79 (100.0%)

Note. Group 1 = Couples who play the same leisure sports together; Group 2 = Couples who play different sports separately; Group 3 = Couples where only one of them plays leisure sports; Group 4 = Couples who do not play leisure sports.

**Table 2 behavsci-14-00757-t002:** Results of factor analysis for stroke.

Items	A	B	C	D	E
I recently received a compliment from my spouse.	0.880	−0.219	−0.178	−0.058	0.282
My spouse compliments me a lot during our marriage.	0.879	−0.183	−0.169	−0.056	0.254
My spouse says “thank you” a lot.	0.873	−0.220	−0.132	−0.069	0.257
I have a spouse who supports me in difficult situations.	0847	−0.201	−0.219	−0.063	0.293
My spouse has a nervous reaction to my criticism.	−0.125	0.860	0.184	0.108	−0.055
I am often criticized by my spouse for my mistakes.	−0.181	0.857	0.165	0.028	−0.120
I have been blamed by my spouse for things that had nothing to do with me.	−0.199	0.849	0.118	0.077	−0.183
I often feel that my spouse is strict.	−0.204	0.836	0.105	0.021	−0.203
I don’t want to argue with my spouse, so I try to avoid it.	−0.159	0.195	0.88	0.094	−0.057
I don’t mind canceling plans with my spouse.	−0.130	0.067	0.874	0.109	−0.129
I feel liberated when I eat alone without my spouse.	−0.181	0.170	0.855	0.063	−0.111
I can’t concentrate on a conversation with my spouse.	−0.114	0.139	0.824	0.180	−0.102
I express my unpleasant feelings to my spouse honestly.	−0.024	0.001	0.034	0.895	−0.076
I tend to be honest about my spouse’s faults.	−0.016	0.062	0.092	0.888	−0.128
I tend to criticize my spouse for misbehavior.	−0.028	0.066	0.145	0.858	−0.069
I get irritated with my spouse when something bad happens.	−0.150	0.095	0.154	0.816	−0.193
I actively help my spouse in times of need.	0.266	−0.165	−0.087	−0.137	0.843
I always remember and celebrate my spouse’s anniversaries.	0.254	−0.176	−0.099	−0.207	0.824
I show appreciation for my spouse’s hard work.	0.247	−0.214	−0.194	−0.121	0.777
I am more active when spending leisure time with my spouse.	0.170	−0.066	−0.074	0.081	0.747
Eigenvalues	8.055	2.841	2.303	1.907	1.213
Variance (%)	40.274	14.206	11.513	9.534	6.063
Cronbach’s alpha	0.966	0.916	0.917	0.905	0.896

Note. A = Positive strokes I receive from my spouse; B = Negative strokes I receive from my spouse; C = No stroke; D = Negative strokes I give to my spouse; E = Positive strokes I give to my spouse.

**Table 3 behavsci-14-00757-t003:** Results of the multivariate analysis of variance.

Variables	Sub-Factors	*df*	*F*	*p*	*η* ^2^	Post-Hoc
Stroke	1. Positive strokes I give to my spouse	3	43.074	<0.001 ***	0.290	a,b > c,d
	2. Positive strokes I receive from my spouse	3	42.049	<0.001 ***	0.285	a > b,c,d
	3. Negative strokes I give to my spouse	3	9.741	<0.001 ***	0.084	a,b,c < d
	4. Negative strokes I receive from my spouse	3	23.725	<0.001 ***	0.183	a < b,c,d
	5. No stroke	3	15.064	<0.001 ***	0.125	a < b,c,d
Marital intimacy		3	42.868	<0.001 ***	0.289	a > b,c,d
Marital satisfaction		3	48.567	<0.001 ***	0.315	a > b,c,d
Divorce intention		3	34.674	<0.001 ***	0.247	a < b,c < d

Note. *** *p* < 0.001.

**Table 4 behavsci-14-00757-t004:** Detailed results of post-hoc analyses.

				Stroke			Marital Intimacy	Marital Satisfaction	Divorce Intention
		1	2	3	4	5			
Group 1	G2	0.003 **	<0.001 ***	0.577	<0.001 ***	<0.001 ***	<0.001 ***	<0.001 ***	<0.001 ***
	G3	<0.001 ***	<0.001 ***	0.169	<0.001 ***	<0.001 ***	<0.001 ***	<0.001 ***	<0.001 ***
	G4	<0.001 ***	<0.001 ***	<0.001 ***	<0.001 ***	<0.001 ***	<0.001 ***	<0.001 ***	<0.001 ***
Group 2	G1	0.003 **	<0.001 ***	0.577	<0.001 ***	<0.001 ***	<0.001 ***	<0.001 ***	<0.001 ***
	G3	<0.001***	0.010 *	0.843	0.118	0.886	0.750	0.936	0.154
	G4	<0.001 ***	0.028 *	0.002 **	0.027 *	0.986	0.412	0.05	<0.001 ***
Group 3	G1	<0.001 ***	<0.001 ***	0.169	<0.001 ***	<0.001 ***	<0.001 ***	<0.001 ***	<0.001 ***
	G2	<0.001 ***	0.010 *	0.843	0.118	0.886	0.75	0.936	0.154
	G4	0.098	0.979	0.056	0.966	0.981	0.963	0.013 *	0.012 *
Group 4	G1	<0.001 ***	<0.001 ***	<0.001 ***	<0.001 ***	<0.001 ***	<0.001 ***	<0.001 ***	<0.001 ***
	G2	<0.001 ***	0.028 *	0.002 **	0.027 *	0.986	0.412	0.05	<0.001 ***
	G3	0.098	0.979	0.056	0.966	0.981	0.963	0.013 *	0.012 *

Note. *** *p* < 0.001, ** *p* < 0.01, * *p* < 0.05; G1 = Couples who play the same leisure sport together; G2 = Couples who play different sports separately; G3 = Couples where only one of them plays leisure sports; G4 = Couples who do not play leisure sports; 1 = Positive strokes I give to my spouse; 2 = Positive strokes I receive from my spouse; 3 = Negative strokes I give to my spouse; 4 = Negative strokes I receive from my spouse; 5 = No stroke.

**Table 5 behavsci-14-00757-t005:** Mean scores of variables by groups.

			Stroke			Marital Intimacy	Marital Satisfaction	Divorce Intention
	1	2	3	4	5			
Group 1	4.0057	4.1121	2.4741	1.9770	1.8420	4.2368	4.0529	1.5996
Group 2	3.5088	3.0794	2.6676	2.5676	2.5471	2.9694	2.8047	2.2000
Group 3	2.9357	2.5179	2.8000	2.9000	2.6643	2.7914	2.9029	2.5571
Group 4	2.5759	2.5918	3.2089	2.9715	2.6013	2.7038	2.3924	3.0823

Note. Group 1 = Couples who play the same leisure sports together; Group 2 = Couples who play different sports separately; Group 3 = Couples where only one of them plays leisure sports; Group 4 = Couples who do not play leisure sports; 1 = Positive strokes I give to my spouse; 2 = Positive strokes I receive from my spouse; 3 = Negative strokes I give to my spouse; 4 = Negative strokes I receive from my spouse; 5 = No stroke.

## Data Availability

The data presented in this study are available on request from the corresponding author.

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
