# Peer review of "Comparative Analysis of Stroke, Marital Intimacy, Marital Satisfaction and Divorce Intention According to the Type of Participation in Marital Leisure Sports Activities"

_behavsci, 2024, doi:10.3390/bs14090757_

Round 1

Reviewer 1 Report

Comments and Suggestions for Authors

This manuscript describes a study of individuals who responded to a survey about their leisure time activities and relationship variables.  The author's used transactional analysis and the concept of stroke to suggest that more positive strokes and fewer negative or no strokes are good for relationship health.  I had several concerns about the manuscript:

1)  I think the theory (TA) needs to be more thoroughly presented.  When I read the title I thought this was about a spouse who had a stroke and leisure activities.  I wonder about changing the title so that readers are not under that same impression.

2) the authors suggested that they surveyed couples, but that wasn't clear in their analysis.  It appeared that they surveyed individuals who were members of a couple.  Thus, what the authors have is one couple member's perspective on both liesure activities and relationship outcomes.

3)  The factor analysis made sense but the no stroke items seemed to indicate disconnection and were not neutral.  How do these items fit with transactional analysis?

4)  When describing the instruments I wanted some sample items, since you list them in a table, it would be great if the authors told the readers that the items would be presented in a table.

5) My biggest concern is that the authors are using "causal" language and putting spending liesure time together before relationship satisfaction.  I would say it could be that people who are more satisfied in their relationship are more likely to spend liesure time togther.  The discussion needs to be toned down to associations not effects. 

6)  I think it is a mistake to suggest that all couples need to find some liesure sport activity to engage in together.  Not all couples are athletic, nor interested in doing a particularly or any sport.  It seems naive to believe that engaging in a sport together would increase relationship satisfaction in distressed relationships.  Competition and criticism could certainly show up when the relationship is already distressed.  Again, authors only have associations that can not be interpreted as causal or directional.       

Comments on the Quality of English Language

Overall the writing itself was understandable.  I think the introduction could be rearranged so that the literatures cited were more organized, such as, 1) explaining TA, 2) discussing what has been found to influence relatoinship satisfaction, 3) discussing what has been found to influence divorce intention, 4) what has been found about how liesure and relationship variables are associated, and whether any of that work was longitudinal,  then explaining the purpose.         

Author Response

Dear Reviewer 1,

Thank you for giving us the opportunity to strengthen our manuscript with your valuable comments and queries. We have worked hard to incorporate your feedback and hope that these revisions persuade you to accept our submission.

1)  I think the theory (TA) needs to be more thoroughly presented.  When I read the title I thought this was about a spouse who had a stroke and leisure activities.  I wonder about changing the title so that readers are not under that same impression.

→ Since this study focuses on 'stroke,' which is a component of TA theory, changing the title may result in losing the essence of the study. Additionally, if further explanation or definition of 'stroke' is needed, I will add that information as well.

2) the authors suggested that they surveyed couples, but that wasn't clear in their analysis.  It appeared that they surveyed individuals who were members of a couple.  Thus, what the authors have is one couple member's perspective on both leisure activities and relationship outcomes.

→ We have changed some words “couples” to “participates who are members of a couple” or “people who are members of a couple” in the 16, 141, and 239-240 lines.

3)  The factor analysis made sense but the no stroke items seemed to indicate disconnection and were not neutral.  How do these items fit with transactional analysis?

→ No stroke is also one of the factors in TA theory, and as mentioned in lines 110-112 of the text, it is related to the concept of accepting and recognizing a partner both verbally and non-verbally. Therefore, no stroke can also be considered related to transactional analysis.

4)  When describing the instruments I wanted some sample items, since you list them in a table, it would be great if the authors told the readers that the items would be presented in a table.

→ We have added this sentence “and detailed information on the items are presented in Table 2.” in this manuscript.

5) My biggest concern is that the authors are using "causal" language and putting spending leisure time together before relationship satisfaction.  I would say it could be that people who are more satisfied in their relationship are more likely to spend leisure time together.  The discussion needs to be toned down to associations not effects. 

→ We have added some associations and changed this sentence to “The significance of the results was confirmed by comparing and analyzing stroke, marital intimacy, marital satisfaction, and divorce intention according to the leisure sports activity type. According to Meunier and Baker [86], it is an undeniable fact that couples with a good relationship feel happy and value the time spent with their spouse. Furthermore, it is also well known that couples with a good relationship engage in leisure activities together more frequently than those who do not have a good relationship. Based on the situation, this study found additional information that doing leisure activities together or doing another type of leisure activities respectively not only had a positive effect on the relationship but also that the effect increased when the couple interacted with each other. Consequently, the result of this study could help couples to improve their quality of life and happiness in married life.”

6)  I think it is a mistake to suggest that all couples need to find some leisure sport activity to engage in together.  Not all couples are athletic, nor interested in doing a particularly or any sport.  It seems naive to believe that engaging in a sport together would increase relationship satisfaction in distressed relationships.  Competition and criticism could certainly show up when the relationship is already distressed.  Again, authors only have associations that cannot be interpreted as causal or directional.   

→ We understood that, and we have deleted the sentence in the manuscript.

Reviewer 2 Report

Comments and Suggestions for Authors

Thank you for the opportunity to read and review this interesting paper. The study explored the strokes, marital intimacy, marital satisfaction, and divorce intentions among couples based on their leisure sports participation. The topic is highly interesting, and based on the previous studies, the authors show clearly that the theme is relevant to study. Findings of the study suggest that leisure sports activities positively affect relationships, and this effect increases when couples participate in the same sport. The findings are quite expected, but especially from Asian viewpoint, they are very interesting and important while women have not typically exercised as much as men or as much as Western women, and marriages are not so equal than in Western countries. I highly recommend and hope these results will be presented as much as possible in different Asian contexts. 

All in all, the manuscript is well written. Background of the study is good, and many-sided enough. Materials and methods are described clearly. Results are presented well. Discussion and suggestions for the further research are relevant. However, I would like to see qualitative study of the same topic in the future. For example, interviews of the couples would give us deeper understanding of the topic. My overall view of the paper is very positive. I have only two critical comments: 

- Table 1. When reporting age, marriage duration etc. are both men and women included? The beginning of the table is little bit unclear, and first I read that these things are reported only among females (because there are lots of stuff under the females and nothing under the males). But by the second reading, I began to think if both sexes are included. 

- Ethical questions of the study should be presented more clearly. I mean for example possible mental harms of the couples or quite limited amount of data. 

I am very happy to see this article published and want to congratulate the authors on the great job. 

Author Response

Dear Reviewer 2,

Thank you for giving us the opportunity to strengthen our manuscript with your valuable comments and queries. We have worked hard to incorporate your feedback and hope that these revisions persuade you to accept our submission.

- Table 1. When reporting age, marriage duration etc. are both men and women included? The beginning of the table is little bit unclear, and first I read that these things are reported only among females (because there are lots of stuff under the females and nothing under the males). But by the second reading, I began to think if both sexes are included. 

→ Yes, we have separated men and women only in the sex section of Table 1, while age, marriage duration, and other factors include data for both men and women.

- Ethical questions of the study should be presented more clearly. I mean for example possible mental harms of the couples or quite limited amount of data. 

→ We have added this sentence “The study participants voluntarily took part in the questionnaire, and there was no potential mental harm to the participants.” in the manuscript.

Round 2

Reviewer 1 Report

Comments and Suggestions for Authors

This version of the manuscript is somewhat improved.  TA is not a common theory used in discussing intimate relationships, at least in the US, so I still think more needs to be explained about the theory itself. 

Although the authors have changed the description of who was surveyed, they still discuss their results as if they surveyed couples.  I think in the last part of the discussion they need to continue to use the phrase "individuals who are part of couple."  I also think they need to only discuss the associations found and not "effects" since this was a cross-section study. 

I also suggest that the authors explain in the text how they formed the groups and the n's of each group as well.   

Author Response

Dear Reviewer 1,

Thank you for giving us the opportunity to strengthen our manuscript with your valuable comments and queries. We have worked hard to incorporate your feedback and hope that these revisions persuade you to accept our submission.

1) This version of the manuscript is somewhat improved.  TA is not a common theory used in discussing intimate relationships, at least in the US, so I still think more needs to be explained about the theory itself. 

> We totally understand your concern. The TA information and examples have been added in lines 68-72.

2) Although the authors have changed the description of who was surveyed, they still discuss their results as if they surveyed couples.  I think in the last part of the discussion they need to continue to use the phrase "individuals who are part of couple."  I also think they need to only discuss the associations found and not "effects" since this was a cross-section study. 

> We have changed the word “couple” to “individuals who are part of a couple” in lines 318 and 324. Moreover, we have changed the sentence that is related to effect to “Based on the situation, this study found additional information that doing leisure activities together or doing another type of leisure activity respectively associated with their psychological factors.” in lines 320-322.

3) I also suggest that the authors explain in the text how they formed the groups and the n's of each group as well.   

> We have added this information in lines 151-154.
